# The Foetal Origins of Allergy and Potential Nutritional Interventions to Prevent Disease

**DOI:** 10.3390/nu14081590

**Published:** 2022-04-12

**Authors:** John O. Warner, Jill Amanda Warner

**Affiliations:** 1National Heart and Lung Institute, Imperial College, London SW3 6LY, UK; 2Paediatric Allergy, Red Cross Memorial Children’s Hospital, University of Cape Town, Cape Town 7700, South Africa; dr.jill.warner@gmail.com

**Keywords:** allergic sensitization, foetal immune ontogeny, gene/environment interactions, asthma, eczema, food allergy, maternal diet, PUFAs, vitamin D, Mediterranean diet, microbiome

## Abstract

The first nine months from conception to birth involves greater changes than at any other time in life, affecting organogenesis, endocrine, metabolic and immune programming. It has led to the concept that the “first 1000 days” from conception to the second birthday are critical in establishing long term health or susceptibility to disease. Immune ontogeny is predominantly complete within that time and is influenced by the maternal genome, health, diet and environment pre-conception and during pregnancy and lactation. Components of the immunological protection of the pregnancy is the generation of Th-2 and T-regulatory cytokines with the consequence that neonatal adaptive responses are also biased towards Th-2 (allergy promoting) and T-regulatory (tolerance promoting) responses. Normally after birth Th-1 activity increases while Th-2 down-regulates and the evolving normal human microbiome likely plays a key role. This in turn will have been affected by maternal health, diet, exposure to antibiotics, mode of delivery, and breast or cow milk formula feeding. Complex gene/environment interactions affect outcomes. Many individual nutrients affect immune mechanisms and variations in levels have been associated with susceptibility to allergic disease. However, intervention trials employing single nutrient supplementation to prevent allergic disease have not achieved the expected outcomes suggested by observational studies. Investigation of overall dietary practices including fresh fruit and vegetables, fish, olive oil, lower meat intake and home cooked foods as seen in the Mediterranean and other healthy diets have been associated with reduced prevalence of allergic disease. This suggests that the “soup” of overall nutrition is more important than individual nutrients and requires further investigation both during pregnancy and after the infant has been weaned. Amongst all the potential factors affecting allergy outcomes, modification of maternal and infant nutrition and the microbiome are easier to employ than changing other aspects of the environment but require large controlled trials before recommending changes to current practice.

## 1. Introduction

During delivery the neonate leaves the controlled and sterile environment of the uterus to be exposed to potentially overwhelming physical challenges. Development through pregnancy can be viewed as a preparation for extra-uterine existence with maternal gene/environment interactions influencing organ development and the programming of foetal immune, metabolic and endocrine responses. This concept arises from the hypothesis that the origins of health or risks of most diseases are a consequence of minor perturbations during organogenesis and periods of rapid cell division. It has evolved into the science known as “The Developmental Origins of Health And Disease (DOHAD)” or in common parlance the first 1000 days from conception to the second birthday [1]. A mismatch between the intra- and extra-uterine environment is more likely to result in adverse outcomes, because programming has not prepared the neonate for different exposures. Birth-cohort and migration studies have highlighted this phenomenon in relation to changing susceptibility to cardio-vascular disease, metabolic syndrome and non-communicable immune mediated diseases [2] (Figure 1). Studies have shown that first-generation people migrating from countries with a low prevalence of allergic disease had a lower prevalence of allergy than second generation immigrants. The younger the child on arrival in a new high allergy prevalence country and the longer duration resident in that country directly correlates with changes in the subsequent incidence of allergic disease. Many factors interact to affect outcomes, but all relate to changes in life style and environmental exposures [3].

The immunological mechanisms underlying allergic disease are the expression of T-lymphocyte-mediated responses to common environmental allergens that are biased towards T lymphocytes with helper-2 (Th-2) activity. Th-2 lymphocytes release peptide regulatory factors (cytokines) such as interleukins (IL) 4, IL-5, IL-9 and IL-13, which trigger the production of immunoglobulin-E (IgE), the allergy promoting antibody, and activate inflammatory cells such as eosinophils. Counter regulation is achieved by T-helper-1 (Th-1) lymphocytes generating cytokines such as interferon-gamma (IFN-gamma) which down-regulates Th-2 activity, while IL-4 suppresses Th-1 function. A group of T-lymphocyte regulators (T-regs) suppress both Th-1 and Th-2 activity by either cell–cell contact or the generation of IL-10 and transforming growth factor (TGF) beta [4]. Based on this paradigm it is likely that either over-expression of Th-2 activity or a failure of control by Th-1 or T-regulatory function will result in a higher probability of the development of allergy and allergic inflammation. Indeed, assaying these counter regulatory factors have been used as biomarkers of allergic disease activity and response to treatment such as to allergen immunotherapy [5]. As Th-1 activity is a feature of response to infection it is of note that the demographic trends in allergic and auto-immune disease prevalence have increased significantly, commensurate with decreases in severe infectious diseases. The basis of the so-called hygiene hypothesis arose from this mechanistic insight. However, it is the overall microbial exposure rather than active infection which affects outcomes and much focus, of late, has been directed to the influence of the human microbio-genome (metagenomics) [6,7].

The pattern of response of T-lymphocytes is dictated by the nature of the signaling from antigen presenting cells (APCs). They, in turn are affected by the nature of the antigen exposure and the presence or absence of co-stimulatory signals. Mucosal epithelial cells are the usual first contact points with antigens, which if recognised as conferring potential danger, by, for instance, having enzymic activity, cytokines known as alarmins (IL-25, 33 and Thymic Stromal Lymphopoietin-TSLP) are released by epithelial cells. The alarmins stimulate innate lymphocytic cells with a type-2 phenotype, which amongst a range of cytokines, release IL-4 and 5. This creates the environment to bias the adaptive response to increased Th-2 activity thereby promoting IgE production and eosinophil activation. This is the normal immune response to parasite infestation but is also characteristic of allergy [8]. APCs generate IL-12, -15, -18 and -23 which predominantly stimulate Th-1 responses while IL-10 from APCs and regulatory T-cells inhibit IL-12 and therefore favour Th-2 activity. Relatively minor perturbations of the balance between Th-1 and Th-2 activity, during primary sensitisation, is likely to have significant effects on outcomes, because the mutual counter regulation of each, orchestrated by IFN-gamma and IL-4 respectively enhances the long-lasting commitment to either Th-1 or Th-2 responses [9].

This review focusses on events during foetal life that influence susceptibility to allergic sensitisation and/or allergic disease. In evaluating the outputs from observational and interventional studies it is important to discriminate between immunological processes leading to allergic sensitisation from those that evolve into the diseases associated with allergy such as eczema, asthma, rhinitis and food allergy. As the modification of maternal and infant diets is easier to achieve than changing other aspects of the early-life environment, the final section focuses on nutrition.

## 2. Immunology of Pregnancy and Neonatal Immune Responses

The neonatal immune system should be able to recognise diverse antigens to identify and destroy pathogens while maintaining tolerance to self, to food, and other harmless environmental exposures. Pregnancy is immunologically equivalent to an organ transplant because the foetus expresses both paternal and maternal antigens to which maternal Th-1 tissue rejecting responses would be expected. Where this occurs, it causes intra-uterine growth retardation or early miscarriage in both murine models and human pregnancies. Counter-balancing this response is in part due to a Th-2 and T-regulatory cytokine profile produced by decidual tissues which down-regulate maternal Th-1 responses (Figure 2) [10]. While several studies support this concept, others are less clear. However, clinical evidence comes from the common observation that Th-1 autoimmune diseases improve during pregnancy [11]. Additional modulation of the maternal Th1 response is affected by the expression of a monomorphic tissue type (HLA-G) on extra-villous trophoblasts at the foeto–maternal interface [12]. Antigen-presenting cells in lymphoid accumulations in the foetal gut express markers of maturation and co-stimulatory molecules from 14–16 weeks’ gestation. They are seen in close apposition with T cells within lymphoid accumulations in the small intestine, which exhibit the capping of receptors, suggesting that primary sensitization has occurred [13].

Newborn infants have allergen reactive T-cells which are characterised by greater production of Th-2 rather than Th-1 cytokines. Normally, the post-natal environment modulates responses with an increase in Th-1 and reduction in Th-2 activity. Infants destined to develop allergic disease have a different pattern with sustained Th-2 activity. There are differences in allergen induced cytokine production at birth in infants who have subsequently developed allergic disease. There is a generally reduced capacity in such infants to generate both IFN-gamma the archetypal Th-1 cytokine and IL-13 a characteristic Th-2 cytokine [14,15]. IL-13 immune reactivity can be detected in the placenta between 16 to 27 weeks’ gestation but not thereafter. From 27 weeks onwards until 34 weeks, IL-13 can be found spontaneously released from foetal mononuclear cells, but then it is only released on cell stimulation [16]. Thus, there must be a very subtle regulation of production of this and likely other cytokines with an interaction between the mother, the placenta and the foetus. Factors that influence the post-natal evolution of the allergic immune response include the route, dose and timing of allergen exposure in association with a disturbed human microbiome (dysbiosis), which forms the basis of the hygiene hypothesis [6]. Many environmental factors which have been linked with rising trends of allergic diseases have embraced the concept of an alteration in the balance of immune responses between those which are associated with an allergic pattern, those associated with protection against infection, and those which regulate all responses (Figure 3).

The neonate has a relatively immature innate immune response to pathogen challenge as a consequence of low circulating complement levels, impaired neutrophil function, natural killer cell function and macrophage activation [17,18]. Adaptive responses are also attenuated and predominantly tolerogenic, thereby reducing the generation of inappropriate responses to harmless antigens during the development of antigen-specific memory. Despite the neonatal immature cellular responses, the foetus can mount an antigen-specific T-cell response to intra-uterine viral infection as early as at 28 weeks’ gestation [18]. Antigen-specific T-cell proliferation has been demonstrated to egg and house-dust mite allergens from 22 weeks’ gestation [19]. Neonates’ relatively inadequate response to infection is partly explained by Th-2 and T-reg cytokines impairing Th-1 responses [17,18].

## 3. Foetal Sensitization to Allergens

Amniotic fluid contains antigens/allergens to which the mother has been exposed together with desidual tissue derived Th-2 cytokines, which are swallowed by the foetus. This is a potential route toTh-2 biased sensitisation in the small intestine [20,21]. A relatively Th-2 biased immune response is well established in murine models but less clear from human studies [22,23]. However, the concept is supported by the positive association between maternal exposure to house dust mite and cat allergen exposure during pregnancy and neonatal cord blood IgE levels [24]. This study also showed a negative correlation between ante-natal endotoxin exposure and neonatal IgE. Endotoxin would be expected to enhance Th-1 responses and suppress Th-2 and therefore restrict IgE production [24]. Furthermore, successive studies have shown that human neonates destined to become allergic have different peripheral blood mononuclear cell (PBMC) responses to allergens [25]. Cord blood PBMCs from neonates born into farming families have high T-regulatory function and lower Th-2 activity in response to house dust mite stimulation. The effect was increased with progressive higher maternal exposure to farm animals and stables [26]. This likely explains the finding of reduced allergy and allergic diseases in the offspring of farming families [27]. However, an alternative factor reducing risks is maternal ingestion of unpasteurized milk which will affect the maternal and infant microbiome and is clearly more likely in faming families and those living in rural environments [28]. In murine models, high house-mite allergen exposure in pregnancy is associated with increased house mite induced bronchial reactivity in offspring [29]. However, house-mite avoidance measures commonly also reduce endotoxin exposures, thereby reducing T-reg and Th-1 activity as suggested by farming studies [26,27,28]. In murine models it is possible to control for all confounding factors which may increase the effect of allergen exposure, but in human observational and interventional studies there are many confounders likely to modify outcomes. It is therefore not surprising that pregnancy house-dust mite avoidance trials have produced no overall effect on allergy, childhood wheezing or asthma [30,31]. Similar, negative outcomes have occurred with trials of pregnancy avoidance of common food allergens. Indeed, avoidance may compromise nutrition [32]. As complete avoidance is rarely achievable and may have adverse effects, high exposure may be a more practical preventive strategy, but controlled trials are required before changing the present advice to continue as normal, but at least there is no justification to avoid any potential allergen for primary prevention. There is likely to be a bell-shaped curve of risk of sensitisation in relation to levels of allergen exposure in pregnancy. Very low levels are insufficient to induce sensitisation while very high levels induce tolerance through a range of mechanisms.

Amniotic fluid also contains IgE antibodies at 10% of maternal circulating levels by 16 weeks’ gestation. IgE receptors are present on cells within the lamina propria of the foetal gut. Therefore, there is the potential for the high-affinity pick-up of antigens by APCs. This is a phenomenon known as antigen focusing. With IgE on low and high affinity IgE receptors, sensitization occurs to 100–1000-fold lower concentrations of allergens than would occur without IgE [33]. While the likely route of primary sensitisation to allergens in pregnancy is via the foetal gut during the second trimester, there is also exposure to allergen directly into the foetal circulation in the third trimester. This is a consequence of the active transport of the IgG antibodies across the placenta complexed with antigens and allergens. Higher IgG antibody levels to specific allergens, due to high maternal exposure, diminish the likelihood of subsequent sensitisation to those allergens. This had been demonstrated for egg, cat and dog allergens [34,35]. Rye-grass allergen immunotherapy continued during pregnancy, which increases IgG antibodies, and has been associated with less subsequent rye-grass allergy in the offspring [36]. Birch and timothy grass pollen exposure via the mother during the first six months of pregnancy increased risks of allergic sensitisation in the infant while later exposure resulted in tolerance in one study [37]. However, a more recent large study showed that higher pollen exposure in late pregnancy was associated with a higher risk of hospital admission for wheezing in infants over the first year of life, while high exposure in the first trimester reduced risks though only in smoking mothers [38]. There was potential for confounding in these studies in relation to maternal vitamin D levels, other potential confounders and post-natal pollutant exposures. [37,38]. There are clearly complex relationships between the timing of exposure to allergens in pregnancy, the concentration, the nature (complexed with IgG or free), co-existent nutritional and other environmental variations all of which have subtly different influences on allergy outcomes.

## 4. Foetal Gene/Environment Interactions

Single nucleotide gene polymorphisms (SNPs) have variously been associated with allergy and/or asthma, and/or eczema, and/or rhinitis, but account for only a small proportion of variability in the phenotype. It has been suggested that it will be more fruitful to focus on gene/environment interactions, epigenetics, and pharmaco-genetics [39]. Twin and family studies have shown that allergy is hereditable, and a many SNPs have been associated with allergy, though not necessarily allergic disease. Numerous mechanistically credible SNPs have been associated with increased risks of allergic sensitization and/or disease, but Variability of phenotyping has compromised the merging of data. Initially, studies utilizing the candidate gene approach focused on the cytokine gene cluster on the long arm of chromosome 5 (5q31–33). This contains the genes for many Th-2 cytokines and an endotoxin receptor, CD-14, which is credible in relation to clinical observations and immunological insights. More recent genome-wide association studies (GWAS) have highlighted many more associations which have begun to separate influences on allergic sensitisation from those which increase risks of specific allergic diseases because the evolution of allergic sensitisation to specific diseases such as eczema or asthma is not inevitable [40]. Indeed, allergic sensitisation sometimes exists in the absence of any disease, and both eczema and asthma occur in the absence of allergic sensitisation. GWAS have highlighted distinct SNPs only expressed on genes in epithelial but not immune cells which increase the risks of eczema (filaggrin SNPs) or asthma (A Dysintegrin And Metalloprotease33—ADAM-33) with or without allergy [41].

However, genetic influences on outcomes contribute relatively small effects because the impact of the polymorphisms will only be apparent at critical stages during development and/or the presence of environmental stressors. For example, ADAM-33 is expressed during airway branching morpho-genesis and SNPs may have their greatest impact very early in foetal life because airway formation is complete by the middle of the second trimester of pregnancy [42]. The metabolism of paracetamol (acetominophen) results in the depletion of anti-oxidant activity, and maternal SNPs in the antioxidant glutathione-S methyl transferase P1 gene only manifests as wheezing in five-year-old offspring if the mother had high usage of paracetamol during pregnancy [43]. Some gene SNPs identified by GWAS affect both airway form and function and immune responsiveness. This is the case for Orosomucoid like 3 (ORMDL3) gene SNPs resulting in over-expression in airway smooth muscle and epithelial cells [44]. Filaggrin SNPs are associated with an increased risk of asthma and food allergy as well as eczema. It is likely that excessive epithelial permeability facilitates sensitisation to allergens through the skin, particularly in the presence of skin inflammation [45] (Figure 4).

The other way in which environment impacts on gene expression is through modification of molecules attached to DNA sequences or the histones around which chromosomes wind. This is the relatively new discipline of epigenetics which not only directly affects phenotype but is also hereditable. Cigarette smoking induces many epigenetic changes and grand-maternal smoking increases risks of asthma in grand-children independent of maternal smoking [46]. Levels of DNA (CpG motif) methylation if increased (hyper-methylation) silences gene expression. Hypo-methylation of the GATA-3 gene in cord blood has been associated with a higher risk of doctor diagnosed asthma at three years of age [47]. Folic acid is a potent methyl donor increasing hyper-methylation and there have been some weak associations between intake of this vitamin in late pregnancy and increased risks of respiratory illness in early childhood [48]. Folic acid supplementation in late pregnancy may increase susceptibility of the infant to respiratory illness, but supplementation pre-conception and in the first half of pregnancy is important to prevent neural tube defects.

## 5. Foetal Nutrition and Allergic Disease

The maternal diet will affect the mother’s gut microbiome and therefore the infant’s inoculum during delivery. However, the maternal microbiome and other confounders have hitherto not been comprehensively analyzed in relation to studies of the impact of pregnancy nutriention on allergy outcomes. This means that outcomes must be interpreted with caution.

Foetal growth and nutrition have an impact on the ontogeny of immune responses. There is an unexpected association between large head circumference at birth and levels of total IgE at birth in childhood and adulthood [49]. It has been hypothesised that a large head circumference at birth is representative of an early rapid foetal growth trajectory because of good nutrient supply in early pregnancy. The foetus is subsequently programmed to maintain a rapid growth trajectory necessitating a high nutrient demand. If this is not met in the later stages of pregnancy there is continuing head growth at the expense of relatively poorer nutrient delivery to the body with consequent adverse effects on immune and lung development. Rapid immediate post-natal weight gain is often a consequence of late intra-uterine growth faltering and is associated with poorer infant lung function [50]. The combination of compromised immune responses and lung function leads to a higher risk of infant wheezing and allergic asthma. High birthweight has been associated with increased risks of subsequently positive allergy skin tests but not necessarily asthma [51]. While allergy has been associated with high foetal abdominal girth growth velocity, infant wheeze was associated with low abdominal growth velocity [52]. These observations are consistent with the concept that allergy is a consequence of affluence and good nutrition while airway disease follows gene/environment aberrations including impaired nutrition, which compromises airway development.

The key question is whether there are any specific nutrients of importance in promoting appropriate immune responsiveness. Reduced intake of fresh fruit and vegetables has been associated with a higher rate of allergic sensitisation in many studies [53]. A high intake of fish in pregnancy has been associated with less subsequent allergy in the offspring [54]. Conversely, a high maternal free sugar intake in pregnancy has been associated with a higher risk of allergy and asthma [55]. A meta-analysis of studies investigating maternal pregnancy intake of vitamins and trace elements suggested a protective effect of vitamins D and E, and zinc against the development of wheezing illnesses in offspring, but inconclusive effects on the prevalence of asthma and other allergic conditions [56]. This emphasizes the distinction between obstructive airway diseases resulting in wheeze and allergic asthma. Conflicting outcomes have been noted for pregnancy intake of numerous other trace elements, many associated with anti-oxidant activity, but heterogeneity of outcome phenotyping precludes meaningful conclusions and association studies do not distinguish cause from consequence [57]. Trials of supplementation in pregnancy are required to establish causal relationships.

Maternal obesity has been associated with higher risks of wheezing illnesses and asthma but not other allergic disorders in children. There are credible mechanistic explanations because mediators, known as adipokines, that are released by adipocytes are pro-inflammatory. They will have effects at the materno-foetal interface leading to foetal immune and neuro-endocrine dysregulation. Obesity also likely has epigenetic effects with enhanced pro-inflammatory gene expression [58]. There are, however, confounding factors affecting outcomes which have not been accounted for in association studies. Overall dietary patterns may be different in obese mothers which could affect constituents implicated as influencing allergy outcomes as discussed above. Maternal obesity increases the risk of caesarean section delivery which in turn results in higher risks of infant asthma and food allergy, probably due to gut dysbiosis [59,60]. A study of elective caesarean section without medical indication showed an association with both infant asthma and allergic rhinitis. Breast feeding, which provides pre-biotic oligosaccharides, abrogated the effect, suggesting that gut dysbiosis was indeed the cause [61]. Pregnancy antibiotics also modify the neonatal microbiome, and infants born to mothers who received antibiotics during pregnancy had increased prevalences of eczema and food allergy [62].

Several studies have focused on lipids as being important in immune ontogeny. Indeed, fatty acids have a crucial role as a source of energy, as the principle component of cell membranes, and as precursors for the synthesis of prostaglandins and leukotrienes. Minor variations in levels could have a profound effect on immune responses. Fish oils have a high level of omega-3 polyunsaturated fatty acids (n-3 PUFAs) and Western diets have a diminished intake of n-3 PUFAs with corresponding increases in n-6 PUFAs. This change has been associated with increasing rates of allergic disease and asthma [63]. Low cord blood n-3:n-6 ratios have been correlated with increased subsequent infant eczema and are related to higher maternal meat eating rather than fish eating [64]. Several randomised controlled studies have employed the administration of a fish oil dietary supplement to mothers through pregnancy and lactation with monitoring of outcomes in the offspring, particularly in high risk cohorts, with conflicting outcomes. Even recent systematic reviews have produced differing results. One found no effect on eczema, wheeze, food allergy or allergic rhinitis, but moderate level evidence of a reduction in egg and peanut allergy in high risk cohorts [65]. This review also showed that probiotic supplementation during the last month of pregnancy and lactation to six months may reduce eczema [65]. However, another systematic review suggested that high dose fish oil supplementation reduced asthma [66]. As a proof of concept that the pregnancy fish oil supplementation and microbiome manipulation has some effects, further studies would be worthwhile

Vitamin D receptors (VDR) have been identified on many immune active cells and vitamin D has important immunoregulatory functions. Most notable are effects on T-regs through TGF-beta expression and signaling. Polymorphisms in the VDR have been linked to an increased risk of asthma [67]. While some studies have associated low vitamin D levels with enhanced inflammation in patients with asthma, others have shown no or negative effects. A systematic review suggests that there is currently no evidence to support supplementation trials in pregnancy [68]. The complexity of VDR function in relation to varying levels of vitamin D exposure and their effects on inflammatory processes requires considerably more elaboration before any clinical implications can be addressed. It is possible that there is a U-shaped curve of vitamin D levels and susceptibility to inflammation with both very high and low levels, increasing risks. Indeed, one study has shown this in relation to IgE levels [69]. We have yet to establish optimal levels for immunological health, which may be different from those for bone health.

Most studies of maternal dietary factors affecting allergic disease outcomes in off-spring have attempted to identify beneficial influences. There are a few recent studies suggesting adverse effects of some nutrients such as refined sugar and pro-inflammatory factors [55,57]. A high intake of red meat increases levels of circulating advanced-glycation end products (AGEs), which are similar to molecules expressed on bacteria. They are recognized by a pattern recognition receptor known as (RAGE) which is particularly expressed in the airway epithelium and activates innate pro-inflammatory responses which are appropriate to handle infection, but a “false alarm” if related to dietary AGEs. Current child high dietary AGE intake has been associated with a high prevalence of asthma [70,71]. However, effects of high AGE intake by pregnant mothers in the one published study to date did not affect child outcomes [72]. A ten-year transgenerational cohort study utilised an early pregnancy dietary inflammatory index (predominantly a high fat intake) and healthy eating index (similar to a Mediterranean diet) to assess the effects on asthma development in children over a nine-year follow-up. A low quality pro-inflammatory diet increased asthma risk, while the health diet was protective [71]. What is unclear is whether the impact on outcomes is exclusively due to either a healthy or unhealthy diet or a balance between each. However, the study of total diet rather than individual nutrients may be more likely to show worthwhile effects.

Associations between omega-3 PUFAs, vitamins D, E and zinc at best show only weak beneficial effects, while intervention trials have produced conflicting results with variable outcomes, phenotyping and confounding being critical issues. As is apparent from all pregnancy gene/environment interactions, the relationships are complex, being influenced by timing, dose, and combinations of exposures which are not normally analysed in observational or intervention studies. Some studies of the so-called Mediterranean diet (Med-Diet) with its high intake of fish, olive oil, fresh vegetables and fruits with low intake of chicken and red-meat (i.e., higher n-3:n-6 ratios) have suggested reductions in some but not all allergic phenotypes. A systematic review indicated that maternal adherence to the Med-Diet reduced offspring wheeze/asthma in the first year of life but not thereafter, and had minor effects on allergy but none on eczema [73]. A more recent birth cohort observational study suggested that the Med-Diet in pregnancy improved offspring small-airway function but not the prevalence of any allergic disease [74]. This would explain the reduction in infant wheeze suggested by the previous review [66]. As the Med-Diet in pregnancy has been shown to be safe and beneficial to both mothers’ and infants’ health, further research into mechanisms and controlled trials are indicated [75]. Principle component analysis of dietary diversity from the UK contingent of the “Prevalence of Infant Food Allergy (PIFA)” study revealed that infants having a more diverse intake of fresh fruit, vegetables and home prepared food had a significantly lower prevalence of food allergy by two years of age, as confirmed by controlled challenge [76]. While the focus of this study was on the diet of infants, it is very likely that the pattern of eating would have been similar for mothers during pregnancy. Therefore, the critical timing for the introduction of a diverse healthy diet remains to be established (Table 1).

## 6. Conclusions

Complex interactions at the materno-placental-foetal interface have a profound influence on the infants’ immune maturation and the likelihood of developing allergic sensitisation and disease. Gene/environment interactions and timing of exposures through pregnancy add further degrees of complexity. Understanding the early life origins of allergy will only be possible by embracing this complexity. Studies will now need to investigate combinations of dietary, pollutant, medication and microbial exposures during pregnancy in relation to genomics, epigenomics, metagenomics and metabolomics in relation to infant/child outcomes. Controlled trials of a “healthy diet” during pregnancy are likely to yield better outcomes than focusing on single nutrients which hitherto have produced disappointing results. The manipulation of the neonates’ evolving microbiome is suggested as another focus for controlled prevention trials.

## Figures and Tables

**Figure 1 nutrients-14-01590-f001:**
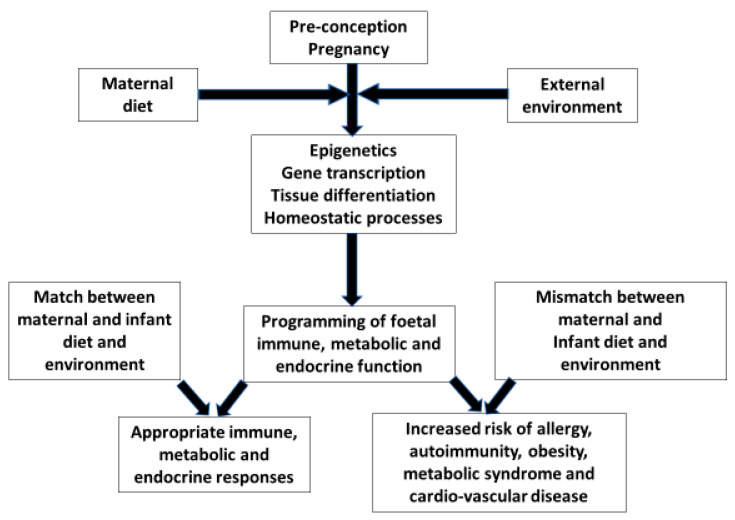
The evolutionary hypothesis based on Crespi BJ Front. Endocrinol. 2020 [2]. The pre-conception and pregnancy maternal gene/environment interactions are the mechanisms by which the foetus is prepared for extra-uterine life. If there is a mismatch between the maternal environment in her earlier life and that of her new-born, programming will be inappropriate for the infants’ environment. This will lead to endocrine, metabolic and immune responses which are ill-equipped to handle environmental exposures with increased risks of cardio-vascular disease, metabolic syndrome and non-communicable inflammatory diseases. Arrows indicate the direction of effect.

**Figure 2 nutrients-14-01590-f002:**
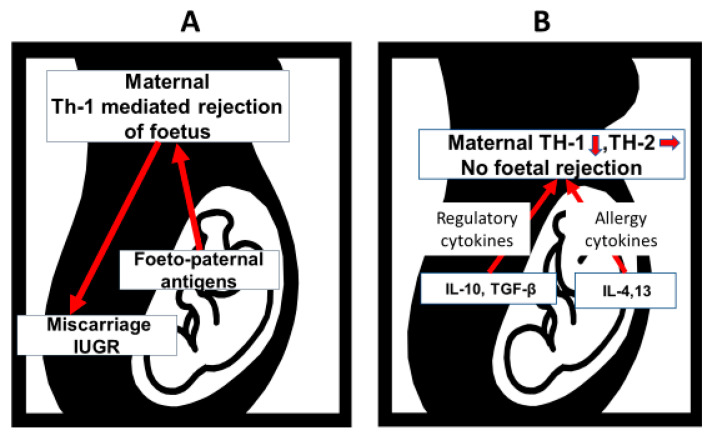
A schematic representation of components of the immunological interactions between mother and foetus. As the foetus and decidual tissues express paternal as well as maternal antigens, a maternal Th-1 tissue rejecting response might be expected, which would compromise the pregnancy with either intra-uterine growth retardation (IUGR) or early miscarriage (**A**). Part of the mechanism to protect the pregnancy is a regulatory and allergy promoting cytokine milieu at the foeto-maternal interface which protects the foetus from maternal Th-1 rejection of foeto-paternal antigens (**B**). The arrows indicate the direction of effect.

**Figure 3 nutrients-14-01590-f003:**
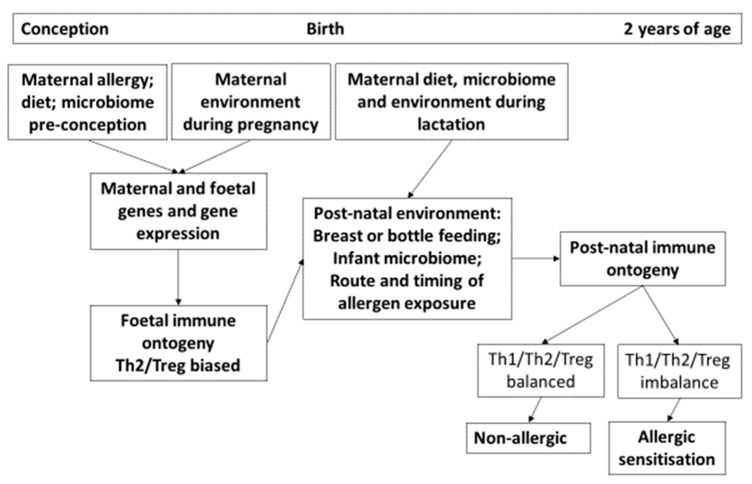
The first 1000 days representing the early life origins of allergic sensitisation. Ante-natal factors include maternal genome, epigenome, metagenome, diet and environment. In the post-natal period, the evolving infant microbiome, diet, route and timing of allergen exposure are critical to balancing the otherwise Th-2 allergy biased neonatal immune response. Failure to down-regulate the neonatal Th-2 biased response is associated with a higher risk of allergic sensitization. The arrows indicate the direction of effect.

**Figure 4 nutrients-14-01590-f004:**
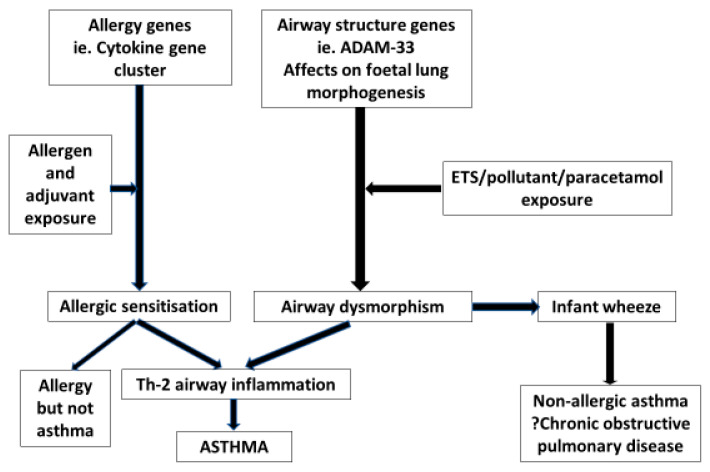
An algorithm of the distinct effects of gene/environment interactions increasing risks of airway dymorphisms and/or allergic sensitisation. When combined they lead to the development of allergic asthma. Isolated airway dysmorphisms increase the risks of infant wheeze and also chronic obstructive pulmonary disease. Arrows indicate the direction of effect.

**Table 1 nutrients-14-01590-t001:** Tabulation of the most promising interventions to prevent allergic disease based on published observational and interventional studies focusing on the first 1000 days. Published evidence referenced in brackets. However, before recommending any interventions it will not only be necessary to demonstrate worthwhile benefit but also to be mindful of potential adverse effects. For instance, paracetamol is the only mild to moderate analgesic recommended for use in pregnancy, while antibiotics for bacterial infection in pregnancy and caesarean section are sometimes essential for medical reasons. Based on all the published evidence, a “healthy diet” would appear to be the best and most practicable option, pre-conception, during pregnancy and lactation.

Timing	Target	Intervention
Pre-conception	Maternal obesity [58]	Weight loss No maternal or grand-mother smoking [46]
Pre-conception	Maternal nutrition	Healthy balanced diet [76]
Pregnancy	Maternal nutrition	More fish less meat Fresh fruit and vegetables [75] Optimal vitamins D, E and zinc [67,68,69,70] No allergen avoidance [29,30,31,32]
Pregnancy	Medications to avoid if possible	Antibiotics [62] Paracetamol [43]
Pregnancy	Maternal microbiome [6]	Pre-/pro-/syn-biotics [6]
Delivery	Avoid if possible	Caesarean section [59,60,61] Bottle feeding [15]
Neonatal period	Infant microbiome [6]	Breast feeding [15] Pre-/pro-/syn-biotics [6]

## Data Availability

Not applicable.

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
