# Peer review of "The Foetal Origins of Allergy and Potential Nutritional Interventions to Prevent Disease"

_nutrients, 2022, doi:10.3390/nu14081590_

Round 1

Reviewer 1 Report

This is a complex topic, and one for which a comprehensive update would be helpful for practitioners. However, I struggled to follow the ‘story’ of the review- many paragraphs jumped from one topic to another and the inflammatory response data often reads like a list of potential cytokines rather than any information that was useful to the reader. Referencing is very poor in this review, especially in the first half- there are many statements that are either un-referenced or given the reference of another review. Those references that are included are surprisingly old-referencing a review from the 1990s does not seem so appropriate to me.

More specific comments:

Within the second half of page 2 and top of page 3 you describe the potential immunological mechanisms of allergic disease. These paragraphs are quite scanty on references. My understanding is that whilst there is fair in vitro evidence for these mechanisms the in vivo work is more controversial. I feel this uncertainty should be more clearly described. Furthermore these paragraphs are scantily referenced- references 3-5 cover this whole area- two of which are reviews, and one of the reviews is 15 years old. I feel this should reference the original work. These paragraphs are also quite hard to follow- a diagram would be very beneficial here- it could be an expansion of figure 3?

On page 4 you state “It is therefore not surprising that numerous pregnancy house-dust mite avoidance trials have produced either no effect, increased or decreased childhood wheezing/asthma (12).” Reference 12 is a review from 2005 that is not specific to pregnancy. It would have been nice to have a more contemporaneous update on this, ideally referencing the primary research.

I am finding some of the paragraphs hard to follow. For example at the base of page 4/top of page 5 you start the paragraph telling us about amniotic fluid IgE, but by the end of the paragraph it seems to be about when antigen in pregnancy antigen exposure occurs. These would be better as separate paragraphs/sections.

“It is possible to detect differences in allergen induced cytokine production at birth in infants who have subsequently developed allergic disease.”- I feel this definitely needs a reference.

The top paragraph on page 6 is also poorly refenced- it only has two references from 1996 and 2010 respectively- what is new since this time?

The middle paragraph of page 6 reads like a list of the findings of reference 22 with little evidence of integration into more contemporaneous literature, or reasoning.

When discussing reference 24 in relation to paracetamol in pregnancy it would definitely be helpful to discuss that this evidence is inconclusive. It almost feels like you are in passing mentioning that paracetamol usage in pregnancy is bad- when for some women this is the only way of them continuing day to day and the risks, benefits and unknowns need to be carefully balanced.

Figure 5 is well explained in the text and one of the highlights of the review. However, the abreviations of GM, M, -ve and + need explanation. Also it would be good to expand upon why the maternal smoking is not postulated to have so much effect on childhood asthma. I also don’t think the effect of folic acid in pregnancy has a place in the same paragraph as this- that could stand alone as its own section.

When discussing omega 3 levels in pregnancy and correlation with offspring health it would be prudent to consider the 2018 Cochrane review on the subject (http://doi.wiley.com/10.1002/14651858.CD003402.pub3) which found that omega 3 supplements reduced the risk of preterm birth- maybe this is the mechanism by which offspring health is improved?

Within table 1 you suggest pre-pro and syn biotics in pregnancy and the neonatal period.  I can’t see discussion of why these are recommended in the text? Table 1 is a nice summary of your findings, but I feel needs references.

Author Response

Thank you for your detailed review.

In order to keep the reference list manageable we used review papers some of which where relatively old but still relevant. We have now expanded the list appreciably and included many more recent references.

Thanks you for pointing out that we jumped between topics. The order has been revised which we hope has improved comprehensibility.

We have justified our statements about mechanisms and slightly modified the emphasis with qualification of in-vitro murine models versus human in-vivo evidence.

We have added qualification to the table in relation to paracetamol and indeed also antibiotics and LSCS in the legend.

We have not added references related to PUFA effects beyond those directly related to allergy. Pre-term birth has no impact on allergy.

Reviewer 2 Report

In this manuscript, Warner & Warner review the interesting topic of fetal origins of allergies. While the authors are field experts and certainly have valuable and thorough knowledge to share with the reader, there are several suggestions to be taken into consideration in an attempt to improve the current version of this submission. Please consider the following suggestions when revising the manuscript:

1- The title is somehow misleading. When reading “Nutrition and the fetal origins of allergic diseases”, one would expect that the article's main focus is on how maternal or infant nutrition affects the development of allergy in the baby, which is not the case here. Please consider changing the title. 

2- The conclusion or the last sentence in this rather wordy abstract is equally misleading. It implies a thorough review of the literature on the impact of nutrition and microbiome on allergies, which is not the case. Please consider either removing this last sentence or changing the manuscript to include more studies on the influence of the microbiome ( please see comments 3). 

3- While the authors repeatedly describe the “evolving normal human microbiome” as a key player, they have only summarized the work of a couple of articles in the field. There are certainly some more studies to include in this area. For instance, the recent review article on “The Infant Microbiome and Its Impact on Development of Food Allergy” by Jungles et al., is definitely worth citing. Similarly for the article by Rachid et al. on ``The microbial origins of food allergy”. 

4- Too many figures ( with low quality), which are crowding the manuscript with no additional value. I suggest deleting some of the figures and replacing the current table with a good-quality figure (for example, please see the figure by the same group in Zepeda-Ortega et al. Front. Immunol., 2021). More specifically, authors could consider the following: 

  • Figure 1 is not self-explanatory. It is unclear what is meant by the mismatch here and this is also not explained in the text
  • Figure 2 looks like a teaching slide rather than a review article illustration
  • Although Figure 3 carries lots of valuable information, it is of mediocre quality and can be greatly improved
  • Same comments for Figure 4, which also appears as a figure from PowerPoint presentation
  • Figure 5 suggest deleting as it is a finding from 1 single paper which is a little unusual to find in a review article
  • Perhaps Table 1 could be replaced by a figure depicting the timeline of neonatal development and the different intervention. 

4- I suggest having a table summarizing all the interesting studies and their findings on the different nutrients influencing infant allergies. 

5- Very few references for a review article. 

  • For instance, the first paragraph in section 3 cites only one reference which describes only the last sentence. 
  • When describing a hot and relevant topic such as the filaggrin mutation, the authors cite only 1 ref; while a PubMed search on the topic yields more than 1000 articles. 
  • The manuscript is lacking recent articles, the authors even fail to cite their own recent work (Zepeda-Ortega et al. Front. Immunol., 2021).

6- Some sentences are ambiguous and need rephrasing. For instance: “Studies have shown that offspring of first-generation migrant women from countries with a low prevalence of allergic disease had a higher prevalence than those delivered by second generation immigrant women”. This sentence is unclear. 

7- There are a few English mistakes that require proofreading, for example:

food allery, dymorphisms.

I hope you can find all the above useful when reviewing the manuscript.

Author Response

Thank you for your review.

We have changed the title with nutrition coming second to foetal immune ontogeny. In addition to nutrition section had been expanded.

We have deleted the original last sentence of the abstract and added an alternative. While we appreciate that microbiome is important and indeed have conducted research and publications in this. We could have added a whole section on microbiome but we are already exceeding the word limit and it is more relevant to post -natal rather than foetal immune ontogeny.

We do not agree with the comments about the figures and tables. Indeed the other reviewer asked for more and commended the table. Teaching slides have considerable value in review articles pitched at a non-tertiary specialist readership. Figure 1 is explained in the first paragraph of the manuscript.

We have increased the reference list significantly as requested by both reviewers. It could of course be very much larger for all the topics discussed in the manuscript but will leave it to the editor to decide whether the current number is adequate. 

Round 2

Reviewer 2 Report

The revised manuscript is a great improvement compared to the previous submission.

No further comments. 

Author Response

Thank you.